# Large magnetoelectric coupling in multiferroic oxide heterostructures assembled via epitaxial lift-off

D. Pesquera [1]✉, E. Khestanova [2], M. Ghidini [1,3,4], S. Zhang [1,5], A. P. Rooney [6], F. Maccherozzi[4], P. Riego[1,7,8], S. Farokhipoor[9], J. Kim[1], X. Moya [1], M. E. Vickers[1], N. A. Stelmashenko [1], S. J. Haigh [6], S. S. Dhesi [4] & N. D. Mathur [1]✉

Epitaxial films may be released from growth substrates and transferred to structurally and chemically incompatible substrates, but epitaxial films of transition metal perovskite oxides have not been transferred to electroactive substrates for voltage control of their myriad functional properties. Here we demonstrate good strain transmission at the incoherent interface between a strain-released film of epitaxially grown ferromagnetic $La_{0.7}Sr_{0.3}MnO_3$ and an electroactive substrate of ferroelectric $0.68Pb(Mg_{1/3}Nb_{2/3})O_3$-$0.32PbTiO_3$ in a different crystallographic orientation. Our strain-mediated magnetoelectric coupling compares well with respect to epitaxial heterostructures, where the epitaxy responsible for strong coupling can degrade film magnetization via strain and dislocations. Moreover, the electrical switching of magnetic anisotropy is repeatable and non-volatile. High-resolution magnetic vector maps reveal that micromagnetic behaviour is governed by electrically controlled strain and cracks in the film. Our demonstration should inspire others to control the physical/chemical properties in strain-released epitaxial oxide films by using electroactive substrates to impart strain via non-epitaxial interfaces.

[1] Department of Materials Science, University of Cambridge, Cambridge CB3 0FS, UK. [2] ITMO University, Saint Petersburg 197101, Russia. [3] Department of Mathematics, Physics and Computer Science, University of Parma, 43124 Parma, Italy. [4] Diamond Light Source, Chilton, Didcot, Oxfordshire OX11 0DE, UK. [5] College of Science, National University of Defense Technology, Changsha 410073, China. [6] School of Materials, University of Manchester, Manchester M13 9PL, UK. [7] CIC nanoGUNE, E-20018 Donostia-San Sebastian, Spain. [8] Department of Condensed Matter Physics, University of the Basque Country, UPV/EHU, E-48080 Bilbao, Spain. [9] Zernike Institute for Advanced Materials, University of Groningen, 9747 AG Groningen, The Netherlands. ✉email: dpesquera@cantab.net; ndm12@cam.ac.uk

Transition metal perovskite oxides display a wide range of functional properties that are mediated by strongly correlated electrons, and thus sensitive to lattice deformations[1,2]. One may therefore strain-tune epitaxial films of these materials according to the choice of single-crystal oxide substrate, e.g. in order to induce structural phase transitions[3], modify charge conduction mechanisms[4,5], enhance ferroic order[6,7] and control chemical reactivity[8,9]. Alternatively, an epitaxial film of any such material can be electrically strained both continuously and discontinuously by a ferroelectric substrate[10]. The continuous response is the well known (converse) piezoelectric effect. The discontinuous response arises due to ferroelectric domain switching, which can sometimes be accompanied by a phase transition[11–13]. Here we will use the term electroactive to collectively describe the continuous and discontinuous responses, thus deviating from the common practice of describing electrically driven strain from ferroelectric substrates purely as piezoelectric. Although electrically driven strain has been widely employed in multiferroic heterostructures for data-storage applications[14–16] based on the electrical control of magnetism[17–19], it could be used more generally to control the various physical and chemical properties and phenomena that arise in transition metal perovskite oxides, as demonstrated for electrical resistivity[20], metal–insulator transitions[21] and photoconductivity[22].

Unfortunately, the key properties of transition metal perovskite oxides are degraded following epitaxial growth on ferroelectric substrates, as seen for ferromagnetic films with reduced Curie temperatures and suppressed magnetizations[17,18,23]. This degradation is a consequence of the >3% lattice-parameter mismatch that causes films to experience both strain and dislocations[17,18,24–26], even if epitaxial buffer layers are present[27]. Moreover, the orientation and magnitude of the voltage-driven in-plane strain[11] is necessarily constrained by the crystallographic orientation required for the epitaxy. Here we solve the mismatch problem, break the orientational constraint and achieve good strain-mediated magnetoelectric coupling by employing epitaxial lift-off[28,29] in order to achieve a form of van der Waals integration[30] via an interfacial glue that formed serendipitously.

Epitaxial lift-off permits an epitaxial film to be transferred to an arbitrary surface after being separated from its growth substrate by chemically etching a sacrificial interlayer. Epitaxial lift-off has been hitherto employed to transfer films of III–V semiconductors[28,29,31,32] and transition metal perovskite oxides[33–40] to passive substrates. Transition metal perovskite oxide films have not been hitherto transferred to electroactive substrates, but one may anticipate viable strain coupling in light of the fact that film properties can be mechanically manipulated via mechanically formed interfaces, as seen for manganite films transferred to flexible substrates[41,42], and two-dimensional structures transferred to electroactive substrates[43].

Our implementation of epitaxial lift-off results in an epitaxially grown film of ferromagnetic $La_{0.7}Sr_{0.3}MnO_3$ (LSMO) transferred to an electroactive single-crystal substrate of $0.68Pb(Mg_{1/3}Nb_{2/3})O_3–0.32PbTiO_3$ (PMN-PT). The LSMO film is grown on (001)-oriented $SrTiO_3$ (STO) via a sacrificial layer of $SrRuO_3$ (SRO), which is subsequently dissolved before transferring the LSMO film to the PMN-PT substrate. We choose STO as the growth substrate because of its small (~1%) lattice mismatch with both LSMO and SRO, and we choose SRO as the sacrificial layer because it can be selectively etched without damaging the LSMO film or STO substrate[44]. We choose the $(001)_{pc}$ orientation of LSMO in order to minimize the magnetocrystalline anisotropy barrier for in-plane rotations of magnetization[45], and we choose the $(011)_{pc}$ orientation of PMN-PT in order to achieve two strain states at electrical remanence[46] (pc denotes pseudocubic).

Ferroelectric domain switching in our PMN-PT substrate is accompanied by a rhombohedral-orthorhombic phase transition, and the large resulting strain is effectively transmitted to the transferred LSMO film, whose magnetization is enhanced after epitaxial lift-off to a value that greatly exceeds the value for LSMO grown directly on PMN-PT[23]. Despite a layer of adsorbates at the interface, the magnetoelectric coupling coefficient is similar to values recorded for epitaxial heterostructures[17,18], and a twofold magnetic anisotropy in the film can be electrically switched by 90° in a repeatable and non-volatile manner. Photoemission electron microscopy (PEEM) with contrast from X-ray magnetic circular dichroism (XMCD) is used to obtain magnetic vector maps that reveal unanticipated complexity, namely few-micron-sized magnetic domains whose boundaries are defined by cracks in the film, and electrically driven magnetic domain rotations through various angles. This complexity represents both a challenge and an opportunity in the development of laminated magnetoelectric memory devices.

## Results

**Sample fabrication and characterization.** An elastomer membrane of polydimethylsiloxane (PDMS) was used to transfer a 45 nm-thick layer of LSMO from its STO (001) growth substrate to platinized PMN-PT $(011)_{pc}$, after dissolving the intervening epitaxial layer of 30-nm-thick SRO with $NaIO_4$ (aq) (ref. [44]) (Fig. 1a). The $a || [100]_{pc}$ and $b || [010]_{pc}$ axes of LSMO that lay parallel to the film edges were approximately aligned with the $x || [100]_{pc}$ and $y || [01\bar{1}]_{pc}$ axes of PMN-PT that lay parallel to the edges of the slightly larger substrate (Supplementary Note 1). For simplicity, samples will be labelled LSMO:PMN-PT, without reference to the Pt electrodes that are considered as if they were an integral part of the PMN-PT substrate, and without reference to an amorphous interfacial layer that we observed in cross-sectional scanning transmission electron microscopy (STEM) (Fig. 1b). Chemical analysis revealed this interfacial layer of adsorbates to be primarily composed of $SiO_x$ and C (Supplementary Note 2), implying partial degradation of the PDMS membrane during the SRO etch. Two other polymers, which might have provided better mechanical support[47], retained only small LSMO flakes after etching.

X-ray diffraction (XRD) measurements of our LSMO/SRO// STO (001) precursor confirmed that the LSMO layer experienced tensile in-plane epitaxial strain of ~1% (red data, Fig. 1c) and a compressive out-of-plane strain of similar magnitude (Supplementary Note 3). The high quality of the LSMO film was confirmed by the presence of thickness fringes, and a narrow $002_{pc}$ rocking curve of full-width half-maximum 0.2° (red data, inset of Fig. 1c). Moreover, XRD reciprocal space maps around the STO 103 reflection confirmed a good match between the in-plane lattice parameters of the LSMO, SRO and STO (Supplementary Note 3). XRD measurements of the LSMO film after it had been transferred to the platinized PMN-PT $(011)_{pc}$ substrate revealed that the epitaxial strain had been completely released (blue data, Fig. 1c), and that the full-width-half-maximum of the $002_{pc}$ rocking curve had increased to 1.7° (blue data, inset of Fig. 1c). This enhancement of texture is attributed to the faceted PMN-PT $(011)_{pc}$ surface that necessarily results from ferroelectric domains of low symmetry (Supplementary Note 4), and cracking in the LSMO film that arose at least in part while it was supported by the flexible PDMS membrane (Supplementary Note 5).

The release of epitaxial strain increased the LSMO saturation magnetization of $2.27 \pm 0.03\ \mu_B$/Mn by 19% to $2.7 \pm 0.1\ \mu_B$/Mn (Fig. 1d), which is roughly six times larger than the room-temperature saturation magnetization of highly strained LSMO grown directly on PMN-PT (ref. [23]). Similarly, the release of epitaxial strain led to an increase of Curie temperature (Supplementary Note 6), as expected[48]. The enhanced magnetism is a

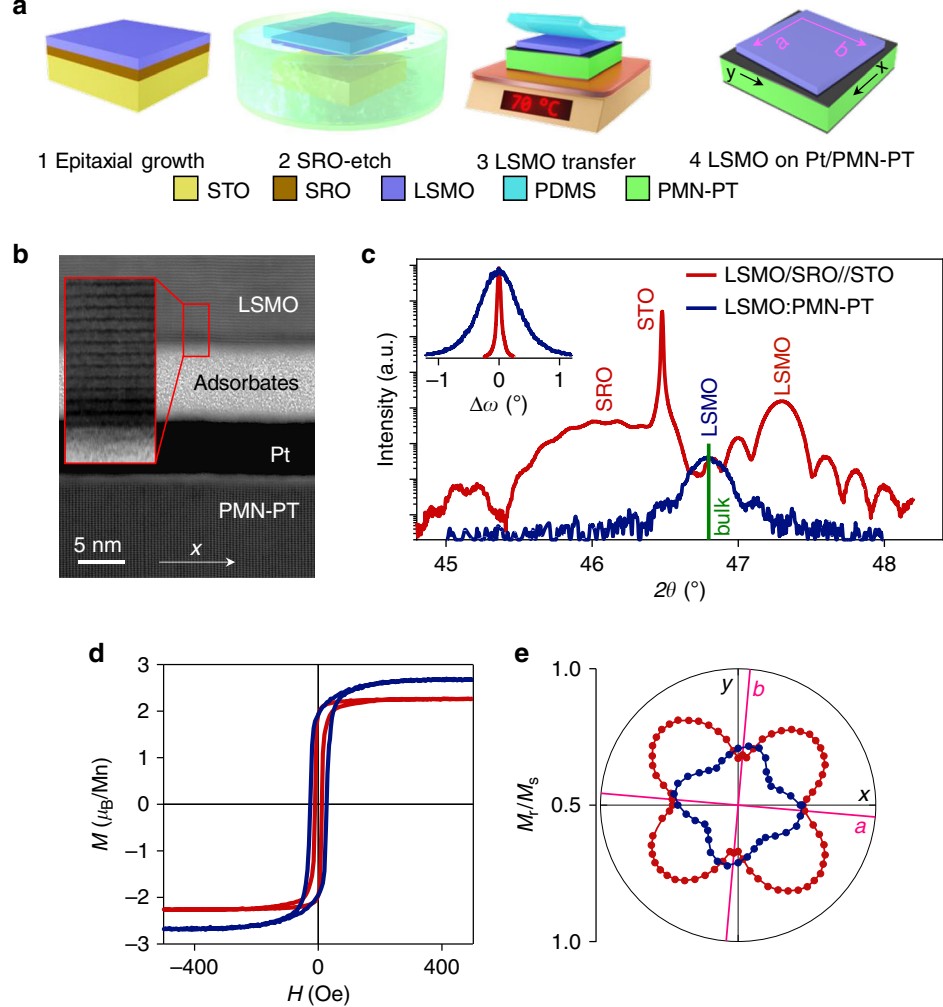

**Fig. 1 Transfer of epitaxial LSMO (001)$_{pc}$ to platinized PMN-PT (011)$_{pc}$. a** The four-step transfer process in which we approximately aligned the edges of the LSMO film (along $a$ || [100]$_{pc}$ and $b$ || [010]$_{pc}$) with the edges of the slightly larger PMN-PT substrate (along $x$ || [100]$_{pc}$ and $y$ || [01$\bar{1}$]$_{pc}$). **b** Bright-field cross-sectional STEM image of the interfacial region between LSMO and PMN-PT, looking down the [01$\bar{1}$]$_{pc}$ zone axis of PMN-PT. The region magnified by ×3.5 confirms in-plane LSMO misalignment. **c–e** Data for electrically virgin LSMO:PMN-PT (blue) and its LSMO/SRO//STO precursor (red). **c** XRD 2$\theta$–$\omega$ scans showing 002$_{pc}$ reflections. The green vertical line corresponds to the expected 002$_{pc}$ reflection for bulk LSMO with pseudocubic lattice parameter[60] 3.881 Å. Inset: 002$_{pc}$ rocking curves for LSMO. **d** Magnetization $M$ versus collinear applied field $H$ for one of the two easy axes. **e** Polar plot of in-plane loop squareness $M_r/M_s$, where $M_r$ denotes remanent magnetization and $M_s$ denotes saturation magnetization. Data in **b** for sample C. Data in **c–e** for sample A and its precursor. All data were obtained at room temperature. Source data are provided as a Source Data file.

consequence of enhanced double exchange following the release of epitaxial strain[49], and should not be attributed to the oxidizing effect of the NaIO$_4$ $_{(aq)}$ reagent on optimally doped films that are well annealed. The release of epitaxial strain also modified the biaxial magnetic anisotropy of the LSMO film (Fig. 1e). After growth, the in-plane LSMO <110>$_{pc}$ directions were magnetically easy due to magnetoelastic anisotropy arising from the biaxial in-plane strain imposed by the STO substrate[45,50,51]. After strain release and film transfer, the in-plane LSMO <100>$_{pc}$ directions were magnetically easy due to uniaxial magnetocrystalline anisotropy in each twin variant of the now twinned film[52] (Supplementary Note 7). Given that the transfer process reduces the magnitude of the easy-axis anisotropy (Fig. 1e), the increase of easy-axis coercivity (Fig. 1d) is inferred to arise because of the observed cracks (Supplementary Note 5) and any accompanying microstructural defects.

**Electrically driven strain in platinized PMN-PT.** After thermally depolarizing PMN-PT in order to set zero strain, a bipolar cycle of

electric field $E$ produced orthogonal in-plane strains $\varepsilon_x$ and $\varepsilon_y$ that took opposite signs to each other at almost every field, and underwent sign reversal near the coercive field (solid butterfly curves, Fig. 2a). Given that the two butterfly curves would be interchanged and different in magnitude if they arose purely from ferroelectric domain switching in rhombohedral PMN-PT of nominally the same composition[46], we infer that polarization reversal was instead associated with a phase transition[11–13], as confirmed by measuring XRD reciprocal space maps while applying an electric field (Supplementary Note 8). Large fields promoted the orthorhombic (O) phase by aligning the polarization along an out-of-plane <011>$_{pc}$ direction, whereas switching through the coercive field promoted rhombohedral (R) twins whose polarizations lay along the subset of <111>$_{pc}$ directions with an out-of-plane component. (A similar argument would hold if this latter phase were monoclinic[53] rather than rhombohedral.)

A minor electrical loop (blue dots in Fig. 2a) permitted two strain states to be created in PMN-PT at electrical remanence[12,46], with $\varepsilon_x = -0.16\%$ in state A and $\varepsilon_x = +0.02\%$

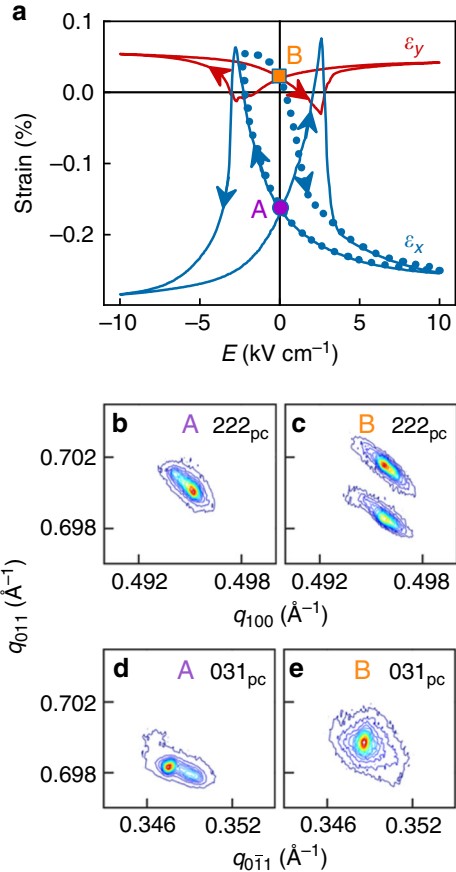

**Fig. 2 Electrically driven strain in platinized PMN-PT (011)$_{pc}$. a** In-plane strain components $\varepsilon_x$ (blue) and $\varepsilon_y$ (red) versus electric field $E$, for PMN-PT with $x \parallel [100]_{pc}$ and $y \parallel [0\bar{1}1]_{pc}$. Remanent states A (purple dot) and B (orange square) were achieved via the minor loop shown for $\varepsilon_x$ (dotted blue line). **b–e** Reciprocal space maps showing the **b, c** 222$_{pc}$ and **d, e** 031$_{pc}$ reflections for PMN-PT at **b, d** A and **c, e** B. Intensity scale runs from purple (low) to red (high). Scattering vector $q = (2/\lambda)\sin(\theta)$ for Bragg angle $\theta$. Data in **a** for annealed PMN-PT from the master substrate used for sample A. Data in **b–e** for sample A. All data were obtained at room temperature. Source data are provided as a Source Data file.

in state B. The corresponding reciprocal space maps obtained at zero electric field (Fig. 2b–e) show for our sample that the O phase dominated in state A (single 222$_{pc}$ reflection, split 031$_{pc}$ reflection), while the R phase dominated in state B (split 222$_{pc}$ reflection, single 031$_{pc}$ reflection). The resulting structural changes in the LSMO film could also be detected by XRD (Supplementary Note 7), despite the twinning and the topography of the underlying ferroelectric domains.

**Strain-mediated electrical control of macroscopic magnetization in LSMO:PMN-PT.** The biaxial magnetic anisotropy that we observed after transfer (Fig. 1e) was rendered uniaxial at both A and B during the course of 30 bipolar electrical cycles (Supplementary Note 9). Subsequent bipolar cycles modified the $x$ and $y$ components of magnetization by ~100% and ~40% (solid butterfly curves, Fig. 3), respectively, and similar results were observed for two similar samples (Supplementary Note 10). The peak magnetoelectric coefficient $\alpha = \mu_0 \mathrm{d}M_x/\mathrm{d}E = 6.4 \times 10^{-8}\,\mathrm{s\,m^{-1}}$ is similar to the value reported[18] for an LSMO film that benefited from good epitaxial coupling with a PMN-PT (001) substrate. The interconversion of remanent states A and B (blue dots in Fig. 3) rotated the single magnetic easy axis by 90° (Fig. 4), while

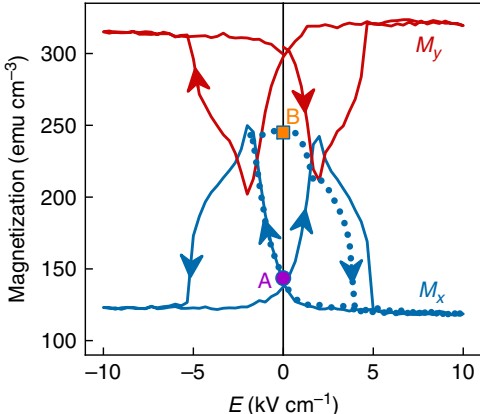

**Fig. 3 Strain-mediated electrical control of macroscopic magnetization in LSMO:PMN-PT.** In-plane magnetization components $M_x$ (blue) and $M_y$ (red) versus electric field $E$. Remanent states A (purple dot) and B (orange square) were achieved via the minor loop shown for $M_x$ (dotted blue line). Data were obtained after applying and removing 1 kOe along the measurement direction, using sample A once it had undergone 30 bipolar electrical cycles (Supplementary Note 9). Similar major loops of $M_x(E)$ were obtained for samples B and C (Supplementary Note 10). All data were obtained at room temperature. Source data are provided as a Source Data file.

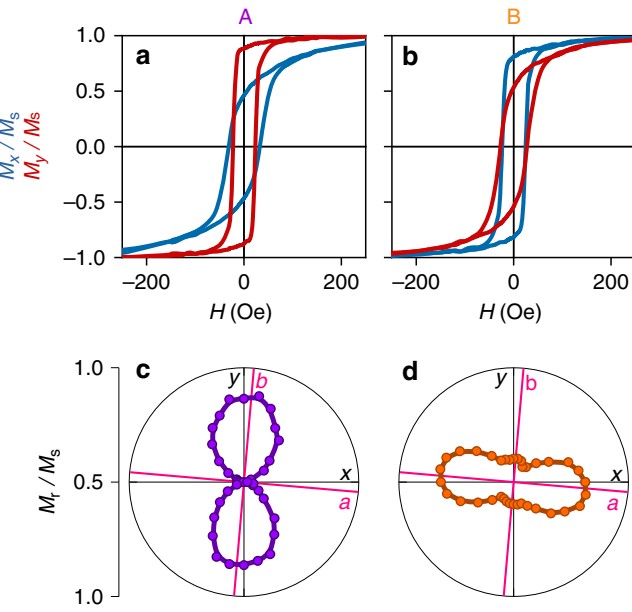

**Fig. 4 Electrically controlled magnetic anisotropy in LSMO:PMN-PT.** For remanent states **a, c** A and **b, d** B, we show **a, b** reduced magnetization components $M_x/M_s$ (blue) and $M_y/M_s$ (red) versus collinear applied field $H$, and **c, d** polar plots of loop squareness $M_r/M_s$ derived from plots that include those shown in **a, b**. Data for sample A after 30 bipolar electrical cycles (Supplementary Note 9). All data were obtained at room temperature. Source data are provided as a Source Data file.

the finite loop-squareness minimum in state B (Fig. 4d) may represent a vestige of the original fourfold anisotropy, or uniaxial regions trapped from state A.

**Strain-mediated electrical control of microscopic magnetization in LSMO:PMN-PT.** The local magnetization of the thermally demagnetized film was imaged using PEEM with contrast

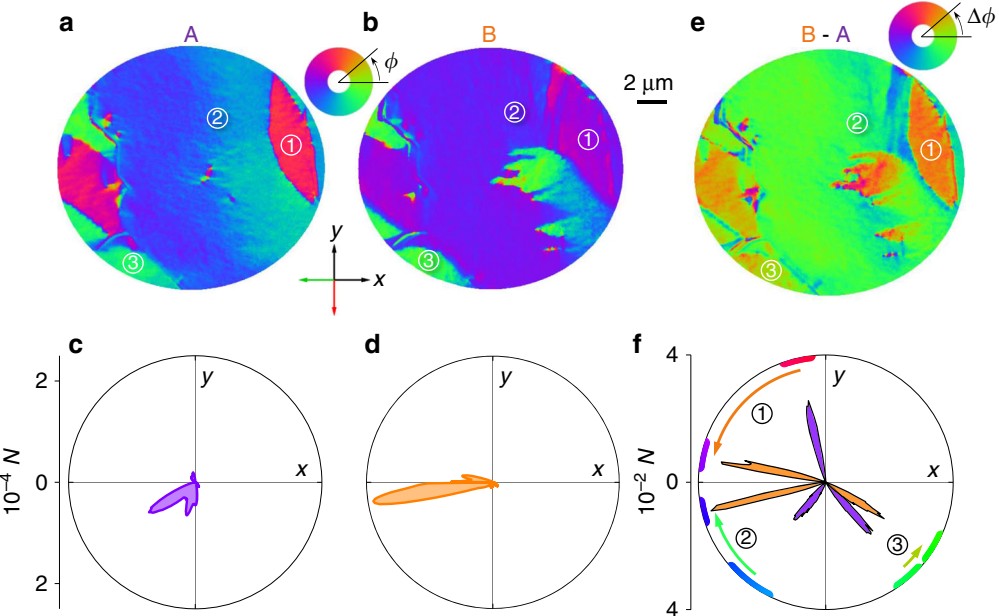

**Fig. 5 Electrically induced rotations of magnetic domains in LSMO:PMN-PT.** For remanent states **a**, **c** A and **b**, **d** B, we show the N pixels in our 20-μm-diameter field of view with in-plane magnetization direction $\phi$ on **a**, **b** XMCD-PEEM vector maps and **c**, **d** polar plots. Comparison of **a**, **b** yields **e** a map of changes in pixel magnetization direction $\Delta\phi$. **f** Polar plot describing 1.3-μm-diameter regions [1–3 in **a**, **b**, **e**] at electrically remanent states A (purple lobes) and B (orange lobes), with pixel colours and angular changes marked on the periphery. Straight green and red arrows denote the in-plane projections of the grazing-incidence X-ray beam that were used to obtain the vector maps in **a**, **b**. All data were obtained for sample A following macroscopic magnetoelectric measurements and subsequent thermal demagnetization. All data were obtained at room temperature. Source data are provided as a Source Data file.

from XMCD. The resulting vector maps of the in-plane magnetization direction $\phi$ revealed that the electrically remanent A and B states were magnetically inhomogeneous within a 20 μm field of view (Fig. 5a, b); and that switching from A to B rotated the net magnetization in our limited field of view towards the x-axis (Fig. 5c, d), consistent with our macroscopic measurements of magnetic anisotropy (Fig. 4c, d). The magnetization was reasonably homogeneous within few-micron-sized domains whose boundaries coincided partly with cracks (Supplementary Note 11), and the electrically driven magnetic rotations in 1.3 μm-diameter regions (1–3 in Fig. 5a, b, e) ranged from large (64° in region 1) to medium (−36° in region 2) to small (16° in region 3) (Fig. 5f). Although the clockwise and anticlockwise nature of these rotations could be explained in terms of the ambipolar shear strains associated with rhombohedral ferroelectric domain switching[54], the very different rotation magnitudes imply the presence of an additional factor. By assuming this additional factor to be a spatially varying uniaxial magnetic anisotropy, due to spatially varying stress associated with the formation of the observed cracks (Supplementary Note 5), a magnetic free energy model was able to approximately reproduce both the local and macroscopic magnetoelectric effects (Supplementary Note 12).

## Discussion

Our macroscopic magnetic measurements yield two key results. First, growth using a well-matched substrate, followed by strain release via epitaxial lift-off[28,29], resulted in an LSMO film magnetization that is greatly enhanced with respect to the value for an LSMO film that contained strain and dislocations following direct growth on PMN-PT[23]. Second, our strain-mediated magnetoelectric coupling is just as effective as strain-mediated coupling across an epitaxial interface[17,18] in spite of the incoherent bonding between our magnetostrictive film and electroactive substrate. Our structurally contiguous micron-sized regions were separated by cracks that could be avoided if one were to transfer

micron-size patterned structures, or modify the procedure and transfer millimetre-sized crack-free films[36]. However, the presence of cracks was instructive because the resulting micromechanical boundary conditions likely served to influence the magnetic rotations of our micron-size magnetic domains.

Close control of these boundary conditions via lithographic patterning would permit a transferred ferromagnetic film to function as the electrically controlled free layer of a magnetic tunnel junction for data-storage applications[14–16], and the free choice of in-plane misorientation angle would permit the realization of schemes for electrically driven magnetization reversal[55–57]. As we have shown, the magnetization would not suffer from the epitaxial suppression that compromises oxide device performance[58]. More generally, the physical and chemical properties of any epitaxially grown film could be electrically controlled via strain after transfer to an electroactive substrate, with no constraints on relative crystallographic orientation, and no epitaxial strain to suppress film functional properties. Separately, it would be interesting to investigate strain-mediated coupling while varying the thickness and composition of the interfacial glue that formed here serendipitously. It would also be interesting to better support the LSMO film during transfer in order to avoid the observed cracks. Ultimately, electroactive substrates themselves could be replaced by electroactive films that have been released in order to avoid substrate clamping[59]. Our work therefore opens the way for multifunctional heterostructures to be assembled from epitaxial oxides via mechanical separation and stacking, just as two-dimensional and other materials may be combined using similar methods of van der Waals integration[30]. These multifunctional heterostructures could then be transferred to silicon host structures in the wider-ranging quest for CMOS compatibility.

## Methods

**Samples.** We fabricated three similar LSMO:PMN-PT samples (A, B and C). The LSMO film (edges along $a \parallel [100]_{pc}$ and $b \parallel [010]_{pc}$) was misaligned with the PMN-

PT substrate (edges along $x \parallel [100]_{pc}$ and $y \parallel [01\bar{1}]_{pc}$) by 5° (samples A, B) and 20° (sample C). All experimental data were obtained using sample A or its precursor components, with the following exceptions: measurements of strain and electrical polarization were obtained using PMN-PT from the same master substrate that we used for sample A; atomic force microscopy (AFM) data were obtained using sample B; STEM data were obtained using sample C; the Curie temperature measurements in Supplementary Fig. 6 were obtained using similar samples; and the macroscopic magnetoelectric data in Supplementary Fig. 10 were obtained using samples A–C.

**Epitaxial growth of LSMO/SRO bilayers**. Epitaxial LSMO (45 nm)/SRO (30 nm) bilayers were grown by pulsed laser deposition (KrF excimer laser, 248 nm, 1 Hz) on STO (001) substrates (5 mm × 5 mm × 1 mm) that had been annealed in flowing oxygen for 90 min at 950 °C. The SRO was grown in 10 Pa $O_2$ at 600 °C (1200 pulses, 1.5 J cm$^{-2}$). The LSMO was grown in 15 Pa $O_2$ at 760 °C (1800 pulses, 2 J cm$^{-2}$). After growth, the LSMO/SRO//STO stacks underwent in situ annealing in 50 kPa $O_2$ for 1 h at 700 °C. Using X-ray reflectivity measurements, the growth rates for single layers of SRO and LSMO were both found to be ~0.025 nm per pulse.

**Platinized PMN-PT substrates**. Each $0.68Pb(Mg_{1/3}Nb_{2/3})O_3–0.32PbTiO_3$ $(011)_{pc}$ substrate (PMN-PT; Atom Optics) was cut to ~5 mm × ~5 mm × 0.3 mm from a different 10 mm × 10 mm × 0.3 mm master. Sputter deposition of Pt resulted in a 6-nm-thick top electrode that served as ground, and a much thicker bottom electrode.

**Transfer of LSMO**. PDMS stamps were cut to 5 mm × 5 mm × 1.5 mm from a commercial specimen (Gelfilm from Gelpak), and each was brought into conformal contact with a given LSMO/SRO//STO stack by heating in air at 70 °C for 10 min (conformal contact was verified by the change in optical reflectance on elimination of the air gap). After floating the resulting PDMS/LSMO/SRO//STO stacks in $NaIO_{4\ (aq)}$ (0.4 M) for several hours, the SRO layers dissolved to release bilayers of PDMS/LSMO, which were washed with deionized water, and dried with $N_2$ gas. Using tweezers, each PDMS/LSMO bilayer was subsequently transferred to a platinized PMN-PT substrate, which had been previously cleaned using acetone and isopropanol, and recently cleaned by annealing in air at 120 °C for 10 min. After transfer, the entire stack was annealed in air (at 100 °C for 10 min) to promote adhesion at the newly formed interface. After cooling to 70 °C and peeling off the PDMS stamp with tweezers, interfacial adhesion was further improved by annealing in air at 150 °C for 10 min.

**X-ray diffraction**. We acquired $2\theta–\omega$ scans and rocking curves for LSMO with a Panalytical Empyrean diffractometer (Cu-K$\alpha_1$, 1.540598 Å), using a hybrid two-bounce primary monochromator on the incident beam, and a two-bounce analyser crystal before the proportional point detector. Reciprocal space maps of PMN-PT were acquired with the same incident beam optics and a PIXcel$^{3D}$ position-sensitive detector, using the frame-based 1D mode with a step time of 10 s.

We used Sample A and its epitaxial precursor to obtain $2\theta–\omega$ scans and rocking curves before applying an electric field (Fig. 1c). For the transferred LSMO film in Sample A, offset angle $\omega$ was obtained by averaging the rocking-curve-peak-values for azimuthal angles of $\varphi = 0°$ and $\varphi = 180°$. Our electric-field-dependent XRD data were also obtained using sample A, after acquiring a subset of the magnetoelectric data and then repeating the last anneal of the fabrication process (10 min in air at 150 °C) in order to depolarize the substrate. We first obtained reciprocal space maps of PMN-PT at successively larger positive fields after negative poling (Supplementary Fig. 5d, e), before acquiring reciprocal space maps for remanent states A and B (Fig. 2b–d). We then obtained $2\theta–\omega$ scans of LSMO for remanent states A and B (Supplementary Fig. 6).

**Atomic force microscopy**. Atomic force microscopy (AFM) images were obtained in tapping mode using a Veeco Digital Instruments Dimension D3100 microscope.

**Electron microscopy**. Cross-sectional transmission electron microscopy (TEM) specimens were prepared via an in situ lift-out procedure in a dual-beam instrument (FEI Nova 600i) that incorporated a focused ion beam microscope and scanning electron microscope in the same chamber. Both 5 and 2 kV ions were used to polish the TEM lamella to a thickness of 50 nm, and remove side damage. High-resolution scanning transmission electron microscopy (STEM) was performed using a probe-side aberration-corrected FEI Titan G2, operated at 80–200 kV with a high-brightness field-emission gun (X-FEG). Bright-field STEM imaging was performed using a probe convergence angle of 21 mrad and a probe current of ~90 pA. In bright-field images, identification of each atomic layer was achieved by elemental analysis using energy dispersive X-ray (EDX) and electron energy loss spectroscopy (EELS). EDX images were obtained using a Super-X four silicon drift EDX detector system with a total-collection solid angle of 0.7 sr. EELS images were obtained using a Gatan Imaging Filter (GIF) Quantum ER system, with an entrance aperture of 5 mm. The lamella was oriented by using the Kikuchi bands to direct the electron beam down the $[01\bar{1}]_{pc}$ zone axis of PMN-PT.

**Strain measurements**. Platinized PMN-PT (derived from the master substrate used for sample A) was cleaned like sample A, using acetone and isopropanol, and then annealed in air at 150 °C for 30 min in order to mimic the final depolarizing heat treatment experienced by sample A. A biaxial strain gauge (KFG-1-120-D16-16 L1M3S, Kyowa) was affixed using glue (CC-33A strain gauge cement, Kyowa) to the top electrode, with measurement axes along $x$ and $y$. The initial values of resistance were used to identify zero strain along the two measurement directions. Strain-field data were obtained while applying bipolar triangular voltages at 0.01 Hz in the range ±10 kV cm$^{-1}$.

**Macroscopic magnetization measurements**. These were performed using a Princeton Measurements Corporation vibrating sample magnetometer, with electrical access to the sample as shown in ref. [17]. All data are presented after subtracting the diamagnetic contribution of substrate, and using optical microscopy to estimate film areas.

**Magnetic vector maps**. After completing all macroscopic magnetoelectric measurements, we obtained raw images of sample A after thermal demagnetization. The electrically remanent states A and B were interconverted in situ using a 300 V power supply that was connected via feedthroughs in the sample holder.

Data were obtained on beamline I06 at Diamond Light Source, where we used an Elmitec SPELEEM-III microscope to map secondary-electron emission arising from circularly polarized X-rays that were incident on the sample surface at a grazing angle of 16°. The probe depth was ~7 nm, and the lateral resolution in our 20 μm-diameter field of view was typically ~50 nm (corresponding to pixels that represent ~20 nm).

Raw images were acquired during 1 s exposure times with right (R) and left (L) circularly polarized light, both on the Mn $L_3$ resonance at 645.5 eV, and off this resonance at 642 eV. The pixels in a raw XMCD-PEEM image describe the XMCD asymmetry $(I^R - I^L)/(I^R + I^L)$, which represents the projection of the local surface magnetization on the incident-beam direction. Here, $I^{R/L} = (I_{on}^{R/L} - I_{off}^{R/L})/I_{off}^{R/L}$ denotes the relative intensity for secondary-electron emission due to X-ray absorption on $(I_{on}^{R/L})$ and off $(I_{off}^{R/L})$ the Mn $L_3$ resonance (the comparison between intensities obtained on and off resonance avoids the influence of any inhomogeneous illumination).

We averaged 40 raw XMCD-PEEM images to obtain a single XMCD-PEEM image for each of two orthogonal sample orientations. These two images were combined in order to yield vector maps of in-plane magnetization, which are not necessarily perfectly circular after correcting for drift and distortion via an affine transformation that was based on topographical images of X-ray absorption for each sample orientation. Each of these topographical images was obtained by averaging all raw images that had been obtained on resonance with left- and right-polarized light.

**XAS images**. X-ray absorption spectroscopy images are presented alongside XMCD-PEEM images (Supplementary Note 11) by plotting $(I^R + I^L)$ and $(I^R - I^L)/(I^R + I^L)$, respectively.

## Data availability

The source data underlying Figs. 1c–e, 2a, 3, 4a–d and 5c–f are provided in a Source Data file. All other relevant data are available from all corresponding authors on request. Source data are provided with this paper.

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

## Acknowledgements

D.P. acknowledges Agència de Gestió d'Ajuts Universitaris i de Recerca (AGAUR) from the Catalan government for Beatriu de Pinós postdoctoral fellowship (2014 BP-A 00079). E.K. acknowledges support by the Ministry of Science and Higher Education of Russian Federation, goszadanie no. 2019-1246. X.M. acknowledges funding from the Royal Society. We thank Diamond Light Source for time on beamline I06 (proposal SI14745-1). S.J.H. and A.P.R. acknowledge funding from EPSRC (Grant EP/P009050/1, EP/M010619/1 and the NoWNano DTC) and the European Research Council (ERC) (ERC-2016-STG-EvoluTEM-715502 and ERC Synergy HETERO2D). P.R. acknowledges funding by the 'la Caixa' Foundation (ID 100010434). We thank Jiamian Hu, Manish Chhowalla, Emilio Artacho, Paul Attfield, Sohini Kar-Narayan and Judith Driscoll for discussions.

## Author contributions

D.P. and N.D.M. designed and led the project. D.P. and E.K. fabricated the samples with assistance from N.A.S. D.P. performed the X-ray diffraction measurements with assistance from M.E.V. A.P.R. and S.J.H. were responsible for the electron microscopy. D.P. acquired the atomic force microscopy data. D.P. performed the electric-field-dependent measurements of magnetization and strain with assistance from M.G. J.K. acquired the temperature-dependent magnetization data. D.P., M.G., F.M., P.R., S.F. and X.M. collected and analysed the XMCD-PEEM data under the supervision of S.S.D. D.P. constructed and analysed the magnetic vector maps. S.Z. performed the magnetic free energy simulations. N.D.M. wrote the manuscript with D.P., using substantive feedback from M.G., and additional feedback from E.K., S.Z., P.R., X.M., M.E.V., S.J.H. and S.S.D.

## Competing interests

The authors declare no competing interests.
