## [Peer Review File · Nature Communications]

Reviewers' comments:

Reviewer #1 (Remarks to the Author):

The authors report on the demonstration of tuning the magnetic properties of LSMO thin films using a single crystal of piezoelectric PMN-PT. The uniqueness of this study is that the LSMO film is not grown epitaxially on the PMN-PT, but grown first on SrRuO₃-templated SrTiO₃ substrate which is subsequently released by selective etching of SrRuO₃ and transferred onto PMN-PT. This method realizes high quality LSMO thin films with lattice constants close to bulk. The strain state of LSMO was successfully manipulated by the applied voltage to the PMN-PT allowing enhancement in bulk magnetization as well as controlling the magnetic anisotropy.

The manuscript is well organized and the benefits of using single crystalline films released from the grown substrate is well presented. Overall, I would like to recommend this manuscript to be published. However, I would like to request the authors to address the following questions:

1. Page 5. The authors state, "...confirmed a good match between all three in-plane lattice parameters." What were the actual lattice parameters? It would be more instructive to show in real numbers.

2. Fig. 1c. Unlike epitaxial LSMO/SRO//STO, in which the substrate 2 θ position can be used as the reference, how accurate is the 2 θ values for the transferred film? The presence of an amorphous layer between Pt and the expected built-in strain gradient in the as-grown LSMO make me think that the LSMO may not be parallel to the substrate surface. Since the authors discuss the structural variation based on XRD with comparison to bulk, characterization of the offset angles along x, y with respect to the PMN-PT substrate would be helpful.

3. I am curious if the as-grown in-plane lattice constants were clamped to the STO substrate or slightly relaxed to the SrRuO₃ in-plane lattice constants since their saturation magnetization of 2.2 $\mu\text{B}/\text{Mn}$ is still below the ideal value of $>\sim 3 \mu\text{B}/\text{Mn}$.

4. Fig. 2(b-e). It would be instructive if positions of ideal diffraction spots are added together with the experimental data. For non-specialists in LSMO or PMN-PT, it would not be easy to capture the relative shift with respect to the expected values.

5. I believe NaIO₄, a very strong oxidant, should be capable of oxidizing LSMO, a mixture of Mn³⁺ and Mn⁴⁺. The enhancement of saturation magnetization, lattice parameter variation, and partial mechanical cracking after release could be partially affected by the (surface) chemical oxidation of LSMO during the selective etching. I am curious if the authors can comment on this point in the text. Possibly, their XAS or EELS show characteristic features of Mn valence in LSMO.

Note: The Supplementary documents were not uploaded and therefore my comments are based on reading the manuscript only.

Reviewer #2 (Remarks to the Author):

Large magnetoelectric coupling in multiferroic oxide heterostructures assembled via epitaxial lift-off

Pesquera et al.

The authors present an experimental study of strain mediated electrical control of the magnetization in

a thin $\text{La}_{0.7}\text{Sr}_{0.3}\text{MnO}_3$ film.

The film was epitaxially grown on $\text{SrRuO}_3/\text{SrTiO}_3(100)$ and transferred, after lift-off, to a $0.68\text{Pb}(\text{Mn}_{1/3}\text{Nb}_{2/3})\text{O}_3-0.32\text{PbTiO}_3$ (PMN-PT) single crystal substrate.

Application of an electric field to the PMN-PT substrate induces 0.1% strain producing strain in the transferred film, enhancing the LSMO magnetization. The magnetic coupling is as strong as in epitaxial LSMO and the magnetization is stronger. The results therefore present a way of circumventing the epitaxial strain-induced suppression of magnetization and suggest that the lift-off technique may be generalizable to other functional perovskite films.

The main result is therefore of interest and sufficiently innovative to merit publication in Nature Communications. However, the manuscript suffers from a number of shortcomings which cast some doubts on the generality of the conclusions and which should be addressed.

The abstract contains the statement that the "strain mediated magnetoelectric coupling is comparable with respect to epitaxial heterostructures in which film magnetization is degraded by epitaxial strain". This requires some explanation, at least in the manuscript, because it is difficult to understand for the more general reader why in the presence of strong coupling the magnetization is degraded. The use of the term "magnetoelectric coupling" is also debatable since in the present case it is the strain which changes the magnetization of the LSMO, not the electric field which is only used to induce either deformation or phase change in the PMN-PT substrate. I would argue that the authors' observation is in some ways closer to magneto-striction than magneto-electric coupling.

It is also difficult to understand why in the introduction (page 3) the authors emphasize ferroelectric substrates when it is the field-induced strain in the substrate which modifies the LSMO magnetization. Presumably, the authors need the ferroelectric switch in PMN-PT to obtain the maximum piezo response? If so this should be clearly stated. Strain could, in principle, also be produced by the converse piezoelectric effect. Is it necessary to switch the ferroelectric polarization to produce the required strain? The authors themselves point out that the strain states are not produced by a straightforward ferroelectric switch in the single rhombohedral phase but rather by the reversible rhombohedral-orthorhombic phase transition during the switch.

Two crucial points in the process are not sufficiently underlined and should be of interest to the general reader. First, that lift-off and transfer allow getting round the inconveniences of the epitaxial constraint, for example, magnetization suppression. Second, transfer allows envisaging in principle integration of perovskite films into CMOS compatible technology, hitherto a real obstacle to applications. Some more discussion of this perspective would be welcome.

Figure 1b shows a 8-9 nm amorphous interface layer (SiO_x and C) formed between the transferred LSMO and the Pt top electrode on the PMN-PT. The effect of this layer is neither investigated nor discussed but it must surely play a key role in that the adhesion of the LSMO to the substrate and therefore its response to strain changes must depend on the coupling through the layer? Indeed, results using other polymers are evoked which suggest that the layer does come from the PDMS membrane. The supplemental information says that weak Van der Waals-like bonding occurs but then it is astonishing that a so weakly bonded layer transmits the high substrate strain to the LSMO. The authors also report the formation of cracks in the transferred LSMO film. The image in the supplemental information suggests that the average crack separation is about 10 micron. Why is there such a dense network of cracks when strain is supposed to be relaxed? When does the cracking happen? Do the cracks evolve as a function of strain? No link made with possible strain release or pinning and the magnetization response of the LSMO. The only discussion on the role of the cracks is some speculation on page 8 about the cracks defining areas of homogeneous magnetization. There is also a reference to extrinsic "enhancement of microstructural complexity" but it is not at all clear to what the authors are referring.

The values for the strain states A and B are also puzzling. The strain along x in state B is 0.02% but in reference 47 the strain is closer to 0.1%, why is this? By the way something has gone wrong with the references, 41 and 47 are identical but with a different author list, I think that ref 47 is the correct list. In Figure 5 the PEEM images are not circular.

My overall impression is that the work is interesting but the manuscript is sometimes superficial and one often has to go into the supplemental information to better understand what is going on. It may be that the manuscript would be better suited to a longer article. The two key points which must be addressed and require more discussion are the interface layer and the cracks in the film.

Reviewer #3 (Remarks to the Author):

In this study, the authors have transferred the epitaxial LSMO membranes on the Pt/PMN-PT substrates and performed an investigation on the multiferroic properties of heterostructures. They have shown an enhancement the saturation magnetization of freestanding LSMO membranes due to the release of epitaxial strain. Furthermore, they observed a large strain-mediated magnetoelectric coupling effect with an incoherent and non-covalent interfacial bonding between ferromagnetic layer and piezoactive substrate. The work is original and can be considered for publication after addressing the following points.

1. The authors mentioned ‘...those transferred to flexible substrates have not been shown to display strain-induced changes of functional properties’, however, in a previous literature, the strain have an effect on the functional properties of the flexible $\text{La}_{0.67}\text{Sr}_{0.33}\text{MnO}_3$ films (ACS Appl. Mater. Interfaces 2019, 11, 22677–22683). This should be clarified.
2. Calculation of the compressive strain in the out-of-plane direction of the LSMO layer in the LSMO/SRO/STO heterostructures seems necessary. Whether the stress in LSMO layer produces the cracks of flexible LSMO? The strain in the LSMO layer will relax by the LSMO thickness. So thicker LSMO layers with less strain will be beneficial to the magnetoelectric coupling effect in the LSMO/Pt/PMN-PT heterostructures.
3. The temperature of the magnetization vs applied field lacks in Figure 1 (d). How about the Curie temperature of LSMO after transfer?
4. In Figure S4, it can be seen that the boundary of the LSMO film during and after transfer is not sharp and a part of the LSMO films fall off. Dose this play a role in the value of the magnetization? Does the direction of cracks in the LSMO membranes change the anisotropy?
5. Could amorphous layer between the LSMO and Pt influence the magnetoelectric coefficient? How to control the thickness of amorphous layer?
6. To better understand the origin of the film microstructure effect, the magnetic properties and magnetoelectric coupling properties with the LSMO grown directly on PMN-PT or the freestanding LSMO membranes free of cracks on PMN-PT, should be shown.
7. There are some typos throughout the reference section that the authors must carefully proof (example – the ref.37).

Reviewer #1

We thank the Referee for the very valuable feedback, which we have used to improve the paper. Changes are highlighted yellow, except for the new Supplementary Notes whose titles are highlighted blue. Changes to Supplementary Note codes and reference codes are not highlighted.

The authors report on the demonstration of tuning the magnetic properties of LSMO thin films using a single crystal of piezoelectric PMN-PT. The uniqueness of this study is that the LSMO film is not grown epitaxially on the PMN-PT, but grown first on SrRuO₃-templated SrTiO₃ substrate which is subsequently released by selective etching of SrRuO₃ and transferred onto PMN-PT. This method realizes high quality LSMO thin films with lattice constants close to bulk. The strain state of LSMO was successfully manipulated by the applied voltage to the PMN-PT allowing enhancement in bulk magnetization as well as controlling the magnetic anisotropy.

The manuscript is well organized and the benefits of using single crystalline films released from the grown substrate is well presented. Overall, I would like to recommend this manuscript to be published.

We thank the referee for these positive comments and for recommending publication.

However, I would like to request the authors to address the following questions:
1. Page 5. The authors state, "...confirmed a good match between all three in-plane lattice parameters." What were the actual lattice parameters? It would be more instructive to show in real numbers.

We have changed:

"XRD reciprocal space maps around the STO 103 reflection (not shown) confirmed a good match between all three in plane lattice parameters"

to:

"XRD reciprocal space maps around the STO 103 reflection confirmed a good match between the in-plane lattice parameters of the LSMO, SRO and STO (Supplementary Note 3)".

and new Supplementary Note 3 shows the reciprocal space map and lattice parameters for the LSMO/SRO//STO (001) sample:

	a (Å)	c (Å)
STO	3.906	3.904
SRO	3.905	3.944
LSMO	3.905	3.841

2. Fig. 1c. Unlike epitaxial LSMO/SRO//STO, in which the substrate 2theta position can be used as the reference, how accurate is the 2theta values for the transferred film? The presence of an amorphous layer between Pt and the expected built-in strain gradient in the as-grown LSMO make me think that the LSMO may not be parallel to the substrate surface. Since the authors discuss the structural variation based on XRD with comparison to bulk, characterization of the offset angles along x, y with respect to the PMN-PT substrate would be helpful.

We now explain in Methods, as we should have originally, that:

“For the transferred LSMO film in Sample A, offset angle ω was obtained by averaging the rocking-curve-peak-values for azimuthal angles of $\varphi = 0^\circ$ and $\varphi = 180^\circ$.”

The rocking curve data for the sample at $\varphi = 0^\circ$ and $\varphi = 180^\circ$ is shown below, and the omega-offset is given by $\omega_{\text{off}} = (\omega_c^{180} + \omega_c^0)/2$.

The offset angles along x and y with respect to the PMN-PT substrate are described in Supplementary Note 1, which is cited from the main text accordingly. However, the offset values only appear in that Note, and we see from the final remark that the Referee was not supplied with this file.

3. I am curious if the as-grown in-plane lattice constants were clamped to the STO substrate or slightly relaxed to the SrRuO3 in-plane lattice constants since their saturation magnetization of 2.2 muB/Mn is still below the ideal value of $>\sim 3$ muB/Mn.

Fig. 1c shows that the out-of-plane pseudocubic lattice parameter of the as-grown LSMO (red peak) is less than the bulk value (green line), and the revision we describe in response to point 1 clarifies that this is indeed a consequence of the in-plane lattice parameter for LSMO being stretched to match the in-plane lattice parameter for STO. As the Referee will know, tensile strain in the LSMO film suppresses magnetization due to the reduced double exchange.

4. Fig. 2(b-e). It would be instructive if positions of ideal diffraction spots are added together with the experimental data. For non-specialists in LSMO or PMN-PT, it would not be easy to capture the relative shift with respect to the expected values.

The purpose of Fig. 2(b-e) is to illustrate peak shifts and splittings in order to identify lattice deformations and phase transformations. It would indeed be good to mark ideal positions, but the PMN-PT spot

positions in Fig. 2(b-e) vary according to stoichiometry and ferroelectric domain configurations. Therefore, precise evaluation of PMN-PT lattice parameters is not possible, and literature values are scattered. We have taken the following two steps to recognise this issue.

1) We have added “for our sample” when discussing Fig. 2(b-e). This change should be read in the context of a change to the previous paragraph, as discussed directly below under point 2).

2) At the start of the section on “Electrically driven strain in PMN-PT”, we had explained that PMN-PT does not always behave in the same way by discussing the signs of the electrically induced strains along x and y , but we did not clarify that we referred to PMN-PT of nominally the same composition (and we did not comment on the magnitudes of these strains). To remedy this, we have added “of nominally the same composition” (and “and different in magnitude”) to the relevant text, which now reads:

“Given that the two butterfly curves would be interchanged and different in magnitude if they arose purely from ferroelectric domain switching in rhombohedral PMN-PT of nominally the same composition⁴⁶, we infer that polarization reversal was instead associated with a phase transition¹¹⁻¹³, as confirmed by measuring XRD reciprocal space maps while applying an electric field (Supplementary Note 8).”

5. I believe NaIO₄, a very strong oxidant, should be capable of oxidizing LSMO, a mixture of Mn³⁺ and Mn⁴⁺. The enhancement of saturation magnetization, lattice parameter variation, and partial mechanical cracking after release could be partially affected by the (surface) chemical oxidation of LSMO during the selective etching. I am curious if the authors can comment on this point in the text. Possibly, their XAS or EELS show characteristic features of Mn valence in LSMO.

We now explain that:

“The enhanced magnetism is a consequence of enhanced double exchange following the release of epitaxial strain⁴⁹, and should not be attributed to the oxidizing effect of the NaIO₄ (aq) reagent on optimally doped films that are well annealed.”

We understand the cracking to arise because the PDMS membrane is flexible, as we now explain at the start of Supplementary Note 5. We now clarify in the paper that we tried and failed to solve this problem with two other polymers, by changing:

“while two other polymers⁴³ that we used for transfer retained only small LSMO flakes after etching”

to:

“Two other polymers, which might have provided better mechanical support⁴⁷, retained only small LSMO flakes after etching.”

Note: The Supplementary documents were not uploaded and therefore my comments are based on reading the manuscript only.

We have asked the Editor to ensure that the supplementary file becomes available for any further rounds of review.

Reviewer #2

We thank the Referee for the very valuable feedback, which we have used to improve the paper. Changes are highlighted yellow, except for the new Supplementary Notes whose titles are highlighted blue. Changes to Supplementary Note codes and reference codes are not highlighted.

The authors present an experimental study of strain mediated electrical control of the magnetization in a thin La_{0.7}Sr_{0.3}MnO₃ film.

The film was epitaxially grown on SrRuO₃/SrTiO₃(100) and transferred, after lift-off, to a 0.68Pb(Mn_{1/3}Nb_{2/3})O₃-0.32PbTiO₃ (PMN-PT) single crystal substrate.

Application of an electric field to the PMN-PT substrate induces 0.1% strain producing strain in the transferred film, enhancing the LSMO magnetization. The magnetic coupling is as strong as in epitaxial LSMO and the magnetization is stronger. The results therefore present a way of circumventing the epitaxial strain-induced suppression of magnetization and suggest that the lift-off technique may be generalizable to other functional perovskite films.

The main result is therefore of interest and sufficiently innovative to merit publication in Nature Communications.

We thank the referee for these positive comments and for recommending publication, subject to the points below.

However, the manuscript suffers from a number of shortcomings which cast some doubts on the generality of the conclusions and which should be addressed.

The abstract contains the statement that the “strain mediated magnetoelectric coupling is comparable with respect to epitaxial heterostructures in which film magnetization is degraded by epitaxial strain”. This requires some explanation, at least in the manuscript, because it is difficult to understand for the more general reader why in the presence of strong coupling the magnetization is degraded.

In epitaxial heterostructures, the epitaxy responsible for the strong coupling can degrade the magnetization because it results in strain and dislocations. To explain this in our abstract, we have changed the quoted sentence to read:

“strain mediated magnetoelectric coupling compares well with respect to epitaxial heterostructures, where the epitaxy responsible for strong coupling can degrade film magnetization via strain and dislocations”.

Note that degradation due to strain and dislocations is explained in paragraph two of the main text, whose presentation has been slightly improved.

In our Discussion, we have slightly improved our explanation about strain and dislocations by changing this:

“epitaxial lift-off^{26,27} greatly enhances LSMO film magnetization with respect to the magnetization of a highly strained LSMO film grown directly on PMN-PT²⁰”

to:

“growth using a well-matched substrate, followed by strain release via epitaxial lift-off^{28,29}, resulted in an LSMO film magnetization that is greatly enhanced with respect to the value for an LSMO film that contained strain and dislocations following direct growth on PMN-PT²³”.

The use of the term “magnetoelectric coupling” is also debatable since in the present case it is the strain which changes the magnetization of the LSMO, not the electric field which is only used to induce either deformation or phase change in the PMN-PT substrate. I would argue that the authors’ observation is in some ways closer to magneto-striction than magneto-electric coupling.

We explain that we have “strain-mediated magnetoelectric coupling” in the abstract and elsewhere, but we did not qualify “magnetoelectric coupling” in the title for simplicity, which is reasonable because (1) the use of “heterostructures” in the title implies an extrinsic coupling and (2) it is standard practice to describe “magnetoelectric coupling” where the system is treated as a black box (e.g. refs 17, 18 and 23).

It is also difficult to understand why in the introduction (page 3) the authors emphasize ferroelectric substrates when it is the field-induced strain in the substrate which modifies the LSMO magnetization. Presumably, the authors need the ferroelectric switch in PMN-PT to obtain the maximum piezo response? If so this should be clearly stated. Strain could, in principle, also be produced by the converse piezoelectric effect. Is it necessary to switch the ferroelectric polarization to produce the required strain? The authors themselves point out that the strain states are not produced by a straightforward ferroelectric switch in the single rhombohedral phase but rather by the reversible rhombohedral-orthorhombic phase transition during the switch.

The Referee has identified a deficiency in our presentation, which arose because the terminology is not clear in the literature. The piezoelectric effect is a continuous effect. There is no special name for the discontinuous changes of strain that arise due to ferroelectric domain switching. The possibility that a phase change accompanies any ferroelectric domain switching is an additional complication. To recognise this problem, we now borrow the term “electroactive” from the language of polymers, to describe most generally an electrically driven strain, whether it be continuous or discontinuous. The resulting changes to the manuscript, described below, will hopefully set a new standard for others to follow.

We explain what we mean by “electroactive” in paragraph 1 of the main text, in revised text that reads as follows:

“an epitaxial film of any such material can be electrically strained both continuously and discontinuously by a ferroelectric substrate¹⁰. The continuous response is the well known (converse) piezoelectric effect. The discontinuous response arises due to ferroelectric domain switching, which can sometimes be accompanied by a phase transition¹¹⁻¹³. Here we will use the term ‘electroactive’ to collectively describe the continuous and discontinuous responses, thus deviating from the common practice of describing electrically driven strain from ferroelectric substrates purely as ‘piezoelectric’.”

Note that “converse” is in brackets because it is common practice to drop this term before “piezoelectric”, and that all but the first ten references are renumbered due to citing what are now refs 11-13 here in respect of the phase transition.

Alas the strict word limit precludes our explanation of ‘electroactive’ from appearing in the abstract, where we now:

- State that “epitaxial films of transition metal perovskite oxides have not been transferred to electroactive substrates for voltage control of their myriad functional properties”.
- Introduce PMN-PT as “an electroactive substrate of ferroelectric $0.68\text{Pb}(\text{Mg}_{1/3}\text{Nb}_{2/3})\text{O}_3\text{-}0.32\text{PbTiO}_3$ ”, where the ferroelectric nature of PMN-PT is useful to state

because PMN-PT is well known as a ferroelectric material, and because the largest changes of strain arise when ferroelectric domains switch, as noted by the Referee.

- Propose future work with “electroactive” not “piezoelectric” substrates.

Elsewhere in the paper we have avoided the incorrect use of “piezoelectric”, including page 3 as noted by the Referee, where we have changed:

“a single crystal substrate of piezoelectric” PMN-PT

to:

“an electroactive single-crystal substrate of” PMN-PT.

Two crucial points in the process are not sufficiently underlined and should be of interest to the general reader. First, that lift-off and transfer allow getting round the inconveniences of the epitaxial constraint, for example, magnetization suppression.

Our abstract explained that we transferred the film to a substrate in a different crystallographic orientation, and that film magnetization is degraded by epitaxial strain. We have now added in the last sentence on future work that epitaxial films would be “strain-released”.

Our final paragraph covered the point by stating what has now been very slightly changed to read:

“As we have shown, the magnetization would not suffer from the epitaxial suppression that compromises oxide device performance⁵⁹. More generally, the physical and chemical properties of any epitaxially grown film could be electrically controlled via strain after transfer to an electroactive substrate, with no constraints on relative crystallographic orientation, and no epitaxial strain to suppress film functional properties.”.

Second, transfer allows envisaging in principle integration of perovskite films into CMOS compatible technology, hitherto a real obstacle to applications. Some more discussion of this perspective would be welcome.

The expanded perspective is an excellent idea, and so we have added this new last sentence:

“These multifunctional heterostructures could then be transferred to silicon host structures in the wider-ranging quest for CMOS compatibility.”.

Figure 1b shows a 8-9 nm amorphous interface layer (SiO_x and C) formed between the transferred LSMO and the Pt top electrode on the PMN-PT. The effect of this layer is neither investigated nor discussed but it must surely play a key role in that the adhesion of the LSMO to the substrate and therefore its response to strain changes must depend on the coupling through the layer? Indeed, results using other polymers are evoked which suggest that the layer does come from the PDMS membrane. The supplemental information says that weak Van der Waals-like bonding occurs but then it is astonishing that a so weakly bonded layer transmits the high substrate strain to the LSMO.

To recognise the role of the interfacial layer, we now write in our introduction that we:

“achieve a form of van der Waals integration³⁰ via an interfacial ‘glue’ that formed serendipitously”.

... and we now write in our final paragraph that:

“it would be interesting to investigate strain-mediated coupling while varying the thickness and composition of the interfacial ‘glue’ that formed here serendipitously.”

To show that good strain transmission across our interface is reasonable, we now present three examples from the literature by explaining in our introduction that:

“one may anticipate viable strain coupling in light of the fact that film properties can be mechanically manipulated via mechanically formed interfaces, as seen for manganite films transferred to flexible substrates^{41,42}, and two-dimensional structures transferred to electroactive substrates⁴³”.

Note that the first of these three new references was brought to our attention by Referee 3.

Note that the incoherent nature of the interface was highlighted explicitly in our abstract, implicitly in our introduction, and explicitly in our Discussion.

The authors also report the formation of cracks in the transferred LSMO film. The image in the supplemental information suggests that the average crack separation is about 10 micron. Why is there such a dense network of cracks when strain is supposed to be relaxed? When does the cracking happen? Do the cracks evolve as a function of strain? No link made with possible strain release or pinning and the magnetization response of the LSMO. The only discussion on the role of the cracks is some speculation on page 8 about the cracks defining areas of homogeneous magnetization. There is also a reference to extrinsic “enhancement of microstructural complexity” but it is not at all clear to what the authors are referring.

We understand that cracks arose because the PDMS membrane that supported the LSMO film during the transfer process is flexible. To explain this, and thus the step at which many cracks were introduced, we have changed:

“cracking in the LSMO film... arose at least in part prior to transfer (Supplementary Note 5)”

to:

“cracking in the LSMO film... arose at least in part while it was supported by the flexible PDMS membrane”.

In Supplementary Note 5, we have deleted the unfounded and perhaps incorrect speculation that cracks are “likely due to epitaxial stress relief after release from the STO substrate”. Instead, we simply note that the PDMS membrane is “flexible”. This is sufficient to briefly convey the point, especially now that we have the aforementioned explanation in the main text.

We did not test whether cracks evolved due to the electrically driven strain, but the key point is that we have cracks after transfer, whether they evolve or not. To explain how these cracks influence the magnetization, we have made the following modifications:

- We have changed:

“the increase of easy-axis coercivity (Fig. 1d) is inferred to arise extrinsically from the enhancement of microstructural complexity”

to:

“the increase of easy-axis coercivity (Fig. 1d) is inferred to arise because of the observed cracks (Supplementary Note 5) and any other microstructural defects”.

... thus deleting our vague description of “microstructural complexity”.

- In Supplementary Note 11 where we use PEEM to show that magnetic domains are bounded by cracks, we have added the suggestion that:
“cracks are at least partially responsible for the coercivity enhancement in the transferred film”.
- The discussion of our model explains that we introduced a spatially varying uniaxial magnetic anisotropy due to stress. We now explain that this stress is “spatially varying”, and that we associate this spatially varying stress “with the formation of the observed cracks (Supplementary Note 5)”.

The values for the strain states A and B are also puzzling. The strain along x in state B is 0.02% but in reference 47 the strain is closer to 0.1%, why is this? By the way something has gone wrong with the references, 41 and 47 are identical but with a different author list, I think that ref 47 is the correct list.

We explained that PMN-PT does not always behave in the same way by discussing the signs of the electrically induced strains along x and y , but we did not comment on the magnitudes of these strains, and we did not clarify that we referred to PMN-PT of nominally the same composition. To remedy this, we have added “and different in magnitude” and “of nominally the same composition” to the relevant text, which now reads:

“Given that the two butterfly curves would be interchanged and different in magnitude if they arose purely from ferroelectric domain switching in rhombohedral PMN-PT of nominally the same composition⁴⁶, we infer that polarization reversal was instead associated with a phase transition¹¹⁻¹³, as confirmed by measuring XRD reciprocal space maps while applying an electric field (Supplementary Note 8).”

We apologise for the mistake of duplicating refs 41 and 47. They are now consolidated as ref. 11, with the correct author list.

In Figure 5 the PEEM images are not circular.

Figure 5 shows vector maps of magnetization that were obtained by combining two XMCD-PEEM images that were obtained with the sample rotated by 90°. This process of combination requires a distortion correction, as explained in Methods, where we have now added that the vector maps “are [therefore] not necessarily perfectly circular”. On balance, we feel that adding this information to the Figure 5 caption would be overly distracting for the majority of readers, who will be able to consult Methods if they are interested to learn more about our vector maps.

My overall impression is that the work is interesting but the manuscript is sometimes superficial and one often has to go into the supplemental information to better understand what is going on. It may be that the manuscript would be better suited to a longer article. The two key points which must be addressed and require more discussion are the interface layer and the cracks in the film.

The supplementary file should be regarded as an integral part of the paper. Including material as supplementary permits the reader to skip details if they are in a hurry, and most people seem to be in a hurry these days. Not, we hasten to add, the Referee whose thoughtful comments are much appreciated.

The issue of the interfacial layer and the cracks are addressed in the specific responses above.

Reviewer #3

We thank the Referee for the very valuable feedback, which we have used to improve the paper. Changes are highlighted yellow, except for the new Supplementary Notes whose titles are highlighted blue. Changes to Supplementary Note codes and reference codes are not highlighted.

In this study, the authors have transferred the epitaxial LSMO membranes on the Pt/PMN-PT substrates and performed an investigation on the multiferroic properties of heterostructures. They have shown an enhancement the saturation magnetization of freestanding LSMO membranes due to the release of epitaxial strain. Furthermore, they observed a large strain-mediated magnetoelectric coupling effect with an incoherent and non-covalent interfacial bonding between ferromagnetic layer and piezoactive substrate. The work is original and can be considered for publication after addressing the following points.

We thank the referee for these positive comments and for recommending publication, subject to the points below.

1. The authors mentioned ‘...those transferred to flexible substrates have not been shown to display strain-induced changes of functional properties’, however, in a previous literature, the strain have an effect on the functional properties of the flexible La_{0.67}Sr_{0.33}MnO₃ films (ACS Appl. Mater. Interfaces 2019, 11, 22677–22683). This should be clarified

We were not aware of this interesting reference, but we have now added it as ref. [41], along with two other related references to have:

“one may anticipate viable strain coupling in light of the fact that film properties can be mechanically manipulated via mechanically formed interfaces, as seen for LSMO films transferred to flexible substrates^{41,42}, and two-dimensional structures transferred to electroactive substrates⁴³.”

We have correspondingly changed the end of the first sentence in our abstract from:

“but epitaxial films of transition metal perovskite oxides have not been transferred to substrates that would permit strain manipulation of their myriad functional properties”

to

“but epitaxial films of transition metal perovskite oxides have not been transferred to electroactive substrates for voltage control of their myriad functional properties”.

2. Calculation of the compressive strain in the out-of-plane direction of the LSMO layer in the LSMO/SRO/STO heterostructures seems necessary. Whether the stress in LSMO layer produces the cracks of flexible LSMO? The strain in the LSMO layer will relax by the LSMO thickness. So thicker LSMO layers with less strain will be beneficial to the magnetoelectric coupling effect in the LSMO/Pt/PMN-PT heterostructures.

Compressive out-of-plane strain in as-grown LSMO

In our introduction, we stated that the lattice mismatch between the LSMO film and its STO growth substrate is ~1%, and in results we report that XRD for the LSMO/SRO//STO heterostructure “confirmed that the LSMO layer experienced tensile in-plane epitaxial strain”. We now show in a new Supplementary Note 3 that the out-of-plane lattice parameter for LSMO in the LSMO/SRO//STO heterostructure is

3.841 Å (Table S1). This is ~1% smaller than the pseudocubic lattice parameter of 3.881 Å for bulk LSMO, which we specify in the caption of Fig. 1. So the ~1% in-plane expansion corresponds to a ~1% out-of-plane compression, as might be expected. We have accordingly changed:

“X-ray diffraction (XRD) measurements of our LSMO/SRO//STO (001) precursor (red data, Fig. 1c) confirmed that the LSMO layer experienced tensile in-plane epitaxial strain.”

to:

“X-ray diffraction (XRD) measurements of our LSMO/SRO//STO (001) precursor confirmed that the LSMO layer experienced tensile in-plane epitaxial strain of ~1% (red data, Fig. 1c) and a compressive out-of-plane strain of similar magnitude (Supplementary Note 3).”

Cracks

We (now) believe that the cracks in the LSMO film do not arise due to the release of epitaxial strain. Instead, we understand that cracks arose because the PDMS membrane that supported the LSMO film during the transfer process is flexible. To explain this, and thus the step at which many cracks were introduced, we have changed:

“cracking in the LSMO film... arose at least in part prior to transfer (Supplementary Note 5)”

to:

“cracking in the LSMO film... arose at least in part while it was supported by the flexible PDMS membrane”.

In Supplementary Note 5, we have deleted the unfounded and perhaps incorrect speculation that cracks are “likely due to epitaxial stress relief after release from the STO substrate”. Instead, we simply note that the PDMS membrane is “flexible”. This is sufficient to briefly convey the point, especially now that we have the aforementioned explanation in the main text.

Thicker layers for improved magnetoelectric coupling

Thicker films of LSMO would be strain relaxed, but as argued above, we believe the cracks in the LSMO film do not arise due to the release of epitaxial strain, so we do not anticipate better coupling with thicker films. More importantly, the whole point of our paper is to demonstrate good magnetoelectric effects with a film that was epitaxially grown without strain relaxation. That said, the strain relaxation in LSMO with an STO substrate would not be as extreme as the strain relaxation in LSMO with a ferroelectric substrate^{17,18}, so thicker films of LSMO would represent an interesting parameter space for future exploration.

3. The temperature of the magnetization vs applied field lacks in Figure 1 (d). How about the Curie temperature of LSMO after transfer?

In fact, the issue of measurement temperature concerns all figures, not just Fig. 1(d). We have therefore added “All data were obtained at room temperature” at the end of every figure caption in the main text. In the Supplementary file, we have noted on the Contents page that “Experimental data in all Notes were obtained at room temperature (except Note 6).”

Plots of remanent magnetization versus temperature in new Supplementary Note 6 show that the Curie temperature of LSMO is enhanced after transfer. This enhancement is due to strain release, as expected from the well-known physics of manganites, explained in new ref. ⁴⁸. Our main text is changed accordingly, such that we now write:

“ the release of epitaxial strain led to an increase of Curie temperature (Supplementary Note 6), as expected⁴⁸.”

4. In Figure S4, it can be seen that the boundary of the LSMO film during and after transfer is not sharp and a part of the LSMO films fall off. Dose this play a role in the value of the magnetization? Does the direction of cracks in the LSMO membranes change the anisotropy?

Magnetization value

We now explain in Methods that the evaluation of film magnetizations involved “using optical microscopy to estimate film areas”.

Anisotropy

We now explain in what has become Supplementary Note 5 that the cracks are “randomly oriented”. Therefore, the cracks should not be responsible for any net anisotropy. However, we now explain that they are associated with a “spatially varying stress” in order to explain the spatially varying uniaxial magnetic anisotropy in our model.

Note that we now explain the related point about how these cracks influence coercivity, by making the following modifications:

- We have changed:
“the increase of easy-axis coercivity (Fig. 1d) is inferred to arise extrinsically from the enhancement of microstructural complexity”
to:
“the increase of easy-axis coercivity (Fig. 1d) is inferred to arise because of the observed cracks (Supplementary Note 5) and any other microstructural defects”.
... thus deleting our vague description of “microstructural complexity”.
- In Supplementary Note 11 where we use PEEM to show that magnetic domains are bounded by cracks, we have added the suggestion that:
“cracks are at least partially responsible for the coercivity enhancement in the transferred film”.

5. Could amorphous layer between the LSMO and Pt influence the magnetoelectric coefficient? How to control the thickness of amorphous layer?

To recognise that details of the interfacial layer might be expected to influence the strain-mediated magnetoelectric coupling, we now write in our introduction that we:

“achieve good strain-mediated magnetoelectric coupling... via an interfacial ‘glue’ that formed serendipitously”.

... and we now write in our final paragraph that:

“it would be interesting to investigate strain-mediated coupling while varying the thickness and composition of the interfacial ‘glue’ that formed here serendipitously.”.

The thickness of the amorphous layer could presumably be controlled by varying the time for which the PDMS is exposed to the $\text{NaIO}_4(\text{aq})$ etch for SRO, as reflected in the fact that we identify “partial degradation of the PDMS membrane during the SRO etch”.

6. To better understand the origin of the film microstructure effect, the magnetic properties and magnetoelectric coupling properties with the LSMO grown directly on PMN-PT or the freestanding LSMO membranes free of cracks on PMN-PT, should be shown.

LSMO grown directly on PMN-PT

We made comparison with this system by citing relevant literature. Specifically, we explain in paragraph 2 of the main text that LSMO grown directly on PMN-PT has degraded properties because it is microstructurally complex due the $>3\%$ lattice-parameter mismatch^{17,18,24–26}. In Results, we explain that our “magnetization was enhanced after epitaxial lift-off to a value that greatly exceeds the value for LSMO grown directly on PMN-PT²³.”

LSMO membranes free of cracks on PMN-PT

To recognise that this would be desirable, we have added in our final paragraph that “It would... be interesting to better support the LSMO film during transfer in order to avoid the observed cracks.” Nevertheless, our key observation transcends the issue of cracks: we have demonstrated good strain-mediated coupling after transferring an epitaxially grown oxide film to an electroactive substrate.

7. There are some typos throughout the reference section that the authors must carefully proof (example – the ref.37).

We thank the Referee for this comment, and we have re-checked all references to ensure accuracy.

REVIEWER COMMENTS

Reviewer #1 (Remarks to the Author):

The authors have fully addressed my questions and concerns. The quality of the manuscript has improved substantially and I recommend publication in Nature Communications.

Reviewer #2 (Remarks to the Author):

Large magnetoelectric coupling in multiferroic oxide heterostructures assembled via epitaxial lift-off

Pesquera et al.

The revised manuscript is much clearer and precise than the first version. The authors clearly state that their results reveal a strain mediated magnetoelectric coupling between an electroactive substrate and strain released epitaxial film of LSMO.

The process is complex and appears to depend on the fortunate but reproducible formation of an amorphous glue transmitting the strain induced by the converse piezoelectric effect in the substrate.

The two strain states in the (011)pc oriented PMN-PT give rise to twofold magnetic anisotropy.

The enhanced magnetism and the increase in the Curie temperature occurs via stronger double exchange. The switching occurs via the structural orthorhombic to rhombohedral phase transition in the PMN-PT substrate.

Thus, the authors present a fascinating and quite complete picture of strain tuning magnetic anisotropy using electrical control of a piezo substrate. The success of the lift-off method should stimulate further research.

The manuscript can be published as it stands. There are just a few minor questions to be addressed/corrected.

On page 5 line 2 when the authors mention the film microstructure they mean (I think) the appearance of cracks. They should say so because otherwise one might think that the film is not a single crystal film.

The role of the amorphous layer in fixing the film and transmitting the substrate strain is well described. The authors should also discuss succinctly how the Pt electrodes might affect the coupling. Pt is neither amorphous nor will the thin Pt electrode have the same strain response as the amorphous "glue".

A minor point on the presentation of the PEEM analysis, it is not necessarily the 20 micron field of view which implies the 50 nm resolution but the overall settings of the microscope and above all the fact that the resolution in PEEM is in any case always poorer than in LEEM (latter uses monochromatic electrons). My guess is that the 50 nm resolution comes from the fact that it is a PEEM experiment and that even with a smaller field of view the spatial resolution would be the same.

The PEEM images are very nice, it is a pity they are not included in the main manuscript but I think there are space limitations.

Reviewer #3 (Remarks to the Author):

I am glad that the authors have settled all doubts of mine in the previous review, thus I recommend the publication.

Reviewer #1 (Remarks to the Author):

The authors have fully addressed my questions and concerns. The quality of the manuscript has improved substantially and I recommend publication in Nature Communications.

We thank the Referee for these positive comments and for recommending publication. Fresh changes are highlighted yellow.

Reviewer #2 (Remarks to the Author):

Large magnetoelectric coupling in multiferroic oxide heterostructures assembled via epitaxial lift-off
Pesquera et al.

The revised manuscript is much clearer and precise than the first version. The authors clearly state that their results reveal a strain mediated magnetoelectric coupling between an electroactive substrate and strain released epitaxial film of LSMO.

The process is complex and appears to depend on the fortunate but reproducible formation of an amorphous glue transmitting the strain induced by the converse piezoelectric effect in the substrate. The two strain states in the (011)pc oriented PMN-PT give rise to twofold magnetic anisotropy. The enhanced magnetism and the increase in the Curie temperature occurs via stronger double exchange. The switching occurs via the structural orthorhombic to rhombohedral phase transition in the PMN-PT substrate.

Thus, the authors present a fascinating and quite complete picture of strain tuning magnetic anisotropy using electrical control of a piezo substrate. The success of the lift-off method should stimulate further research.

The manuscript can be published as it stands.

We thank the Referee for the positive comments, for the detailed feedback that we have used to improve our paper, and for recommending publication. Fresh changes are highlighted yellow.

There are just a few minor questions to be addressed/corrected.

On page 5 line 2 when the authors mention the film microstructure they mean (I think) the appearance of cracks. They should say so because otherwise one might think that the film is not a single crystal film.

Yes, we mistakenly used ‘microstructure’ purely to describe cracks, and so we have changed:

“...boundaries were defined by the film microstructure”

to:

“...boundaries were defined by cracks in the film”

The role of the amorphous layer in fixing the film and transmitting the substrate strain is well described. The authors should also discuss succinctly how the Pt electrodes might affect the coupling.

Pt is neither amorphous nor will the thin Pt electrode have the same strain response as the amorphous “glue”.

The strain gauge measures the PMN-PT substrate in the presence of the Pt electrodes that are used to drive piezoelectric effects, and therefore it is reasonable to consider the Pt electrodes as if they were an integral part of the substrate. To recognise this matter, we have:

- Changed:

“For simplicity, samples will be labelled LSMO:PMN PT, without reference to the Pt electrodes on either side of the PMN PT substrate”

to:

“For simplicity, samples will be labelled LSMO:PMN PT, without reference to the Pt electrodes that are considered as if they were an integral part of PMN PT substrate”.

- Changed the section heading “Electrically driven strain in PMN-PT” to “Electrically driven strain in PMN-PT”.
- Changed the Fig. 2 title “Electrically driven strain in PMN-PT (011)_{pc}” to “Electrically driven strain in platinized PMN-PT (011)_{pc}”.

A minor point on the presentation of the PEEM analysis, it is not necessarily the 20 micron field of view which implies the 50 nm resolution but the overall settings of the microscope and above all the fact that the resolution in PEEM is in any case always poorer than in LEEM (latter uses monochromatic electrons). My guess is that the 50 nm resolution comes from the fact that it is a PEEM experiment and that even with a smaller field of view the spatial resolution would be the same.

The field of view influences resolution via pixel size, but the field of view does not uniquely determine resolution as pointed out by the Referee. To recognise this, we now specify resolution neutrally with respect to the field of view, and link resolution to pixel size. Our text in Methods is thus changed from:

“the 20 μm diameter field of view that we used in this study implies a lateral resolution of ~50 nm, with each pixel representing ~20 nm”

to:

“the lateral resolution in our 20 μm diameter field of view was typically ~50 nm (corresponding to pixels that represent ~20 nm)”

The PEEM images are very nice, it is a pity they are not included in the main manuscript but I think there are space limitations.

The vector maps that we construct from PEEM images appear in Fig. 5 of the main paper. Note that the only PEEM images in the supplementary file show that magnetic domains correlate with cracks that we observed in XAS images (Fig. S11).

Reviewer #3 (Remarks to the Author):

I am glad that the authors have settled all doubts of mine in the previous review, thus I recommend the publication.

We thank the Referee for this positive comment and for recommending publication. Fresh changes are highlighted yellow.

REVIEWERS' COMMENTS:

Reviewer #2 (Remarks to the Author):

The last few details have been clarified and the manuscript is now fully suitable for publication.

REVIEWERS' COMMENTS:

Reviewer #2 (Remarks to the Author):

The last few details have been clarified and the manuscript is now fully suitable for publication.

We thank the Referee for recommending publication.